# Long-Term Carbapenems Antimicrobial Stewardship Program

**DOI:** 10.3390/antibiotics10010015

**Published:** 2020-12-26

**Authors:** José Francisco García-Rodríguez, Belén Bardán-García, Pedro Miguel Juiz-González, Laura Vilariño-Maneiro, Hortensia Álvarez-Díaz, Ana Mariño-Callejo

**Affiliations:** 1Infectious Diseases Unit, Department of Internal Medicine, University Hospital of Ferrol, Sergas, 15405 Ferrol, Spain; laura.vilarino.maneiro@sergas.es (L.V.-M.); hortensia.alvarez.diaz@sergas.es (H.Á.-D.); ana.marino.callejo@sergas.es (A.M.-C.); 2Department of Pharmacy, University Hospital of Ferrol, Sergas, 15405 Ferrol, Spain; belen.bardan.garcia@sergas.es; 3Department of Microbiology, University Hospital of Ferrol, Sergas, 15405 Ferrol, Spain; pedro.miguel.juiz.gonzalez@sergas.es

**Keywords:** antimicrobial stewardship, multidrug-resistant, hospital infections, bloodstream infections, carbapenems, candidemia

## Abstract

Objective. To evaluate clinical and antibiotic resistance impact of carbapenems stewardship programs. Methods: descriptive study, pre-post-intervention, between January 2012 and December 2019; 350-bed teaching hospital. Prospective audit and feedback to prescribers was carried out between January 2015 and December 2019. We evaluate adequacy of carbapenems prescription to local guidelines and compare results between cases with accepted or rejected intervention. Analysis of antibiotic-consumption and hospital-acquired multidrug-resistant (MDR) bloodstream infections (BSIs) was performed. Results: 1432 patients were followed. Adequacy of carbapenems prescription improved from 49.7% in 2015 to 80.9% in 2019 (*p* < 0.001). Interventions on prescription were performed in 448 (31.3%) patients without carbapenem-justified treatment, in 371 intervention was accepted, in 77 it was not. Intervention acceptance was associated with shorter duration of all antibiotic treatment and inpatient days (*p* < 0.05), without differences in outcome. During the period 2015–2019, compared with 2012–2014, decreased meropenem consumption (Rate Ratio 0.58; 95%CI: 0.55–0.63), candidemia and hospital-acquired MDR BSIs rate (RR 0.62; 95%CI: 0.41–0.92, *p* = 0.02), and increased cefepime (RR 2; 95%CI: 1.77–2.26) and piperacillin-tazobactam consumption (RR 1.17; 95%CI: 1.11–1.24), *p* < 0.001. Conclusions: the decrease and better use of carbapenems achieved could have clinical and ecological impact over five years, reduce inpatient days, hospital-acquired MDR BSIs, and candidemia, despite the increase in other antibiotic-consumption.

## 1. Introduction

Antimicrobial resistance is a threat to global public health and many countries reaffirm their commitment to develop national action plans to deal with the problem, establishing, among other measures, systems to guarantee a more appropriate use of antibiotics [1].

Antimicrobial overuse and inappropriate use remain significant problems and, although the highest consumption of antibiotics is at the outpatient level, most studies on the impact of antimicrobial stewardship programs (ASP) focused on the hospital setting, where patients are subjected to a higher pressure of antibiotic treatment and there is a greater risk of transmission of resistance.

Antimicrobial stewardship are quality improvement programs that include heterogeneous interventions [2]. The implementation of these ASP has shown a significant reduction of antibiotic use and hospital costs [3], but few studies refer to the impact on clinical outcome [4], antibiotic resistance [5,6], or incidence of *Clostridioides difficile* infection [7]. Interventions are generally more effective in prospective studies with clinical feedback on prescribing, but there are few studies of this type that cover the entire hospital with a duration long enough in time to evaluate the persistence of their effect [8,9].

The first steps toward ASP in an organization are to identify a leader or leaders for the program and to allocate sufficient administrative support for the ASP [10]. Limited resources for the antimicrobial stewardship implementation make necessary prioritizing those interventions that might have greater impact. Any antibiotic can have a potential effect on the development of bacterial resistance, but that effect and the persistence of resistance over time depends on the type of antibiotic and bacterial species, which may be more or less easy to develop resistance, and may have different rates of transmission of resistance [11,12,13]. In addition to the impact on bacterial resistance, different antimicrobial agents have a different impact on the human gastrointestinal colonization by the *Candida* species, depending on the pharmacological and antibacterial properties of each drug. It has been demonstrated in the murine model that carbapenems cause high increase of yeast counts in the gastrointestinal tract [14], and the use of carbapenems in the treatment of hospitalized patients is associated with the development of candidemia [15].

The increasing number of infections caused by extended spectrum ß-lactamase (ESBL) producing *Enterobacteriaceae* has led to an increase in carbapenems consumption, and the appearance and spread of carbapenem-resistant enterobacteriaceae, causing infections with high mortality due to the shortage of treatment alternatives [16].

Carbapenem-resistant enterobacteriaceae rank among the top three multidrug-resistant pathogens on the World Health Organization (WHO)’s priority list [17]. The aim of the study is to evaluate the clinical and antibiotic resistance impact of carbapenems stewardship implementation over 5 years. This manuscript is an extension of a previously published partial one [18], expanding the analysis to all carbapenems and adding years of study.

## 2. Results

### 2.1. Adequacy of Treatment

Between 2015 and2019, 1432 patients received treatment with carbapenems (1280 meropenem and 152 ertapenem). The indication of justified treatment with carbapenems progressively improved over time from 49.7% in 2015 to 80.9% in 2019, *p* <0.001.

### 2.2. Clinical Impact

Between 2015 and 2019, 1432 patients received treatment with carbapenems, of which 920 were male and 512 female; age 69.7 ± 15.2 years (range 1–97 years). The sites of infection were: urinary 518 (36.2%), abdominal 408 (28.5%), pulmonary 315 (22%), skin and soft tissue 60 (4.2%), febrile neutropenia 31 (2.2%), intravascular catheter 28 (1.9%), other 72 (5%). Place of infection acquisition: hospital-acquired 635 (44.3%), healthcare-associated 559 (39%), and community-associated 238 (16.6%).

Out of the 1432 patients who received treatment with carbapenems, in 984 (68.7%) of them the treatment was considered justified; in 448 (31.3%) the treatment was not considered justified and interventions were performed with a suggestion for appropriate treatment: in 371 (83%), the intervention was accepted, and in 77 it was not.

The clinical characteristics of patients were similar between patients with and without acceptance of ASP recommendations, although the degree of intervention acceptance varied according to prescriber (between 29% and 100%) and infection localization (Table 1).

There were no significant differences between cases with accepted intervention and cases with rejected intervention in clinical outcome or collateral damage. The acceptance of the intervention was associated with shorter duration inpatient days (*p* < 0.05), and less development of yeast colonization or infection, but not statistically significant (Table 2).

The Charlson index was similar throughout the intervention period (5.4 in 2015 vs. 5.8 in 2019, *p* = 0.07) and was higher in patients who died: 7.1 ± 2.3 vs. 5.3 ± 2.6, *p* < 0.001. The duration of all antibiotic treatment along the series decreased significantly from 2015 (12.8 ± 11.3 days) to 2019 (10.8 ± 10.1), *p* = 0.03.

Coinciding with the start-up of ASP, a significant change point was observed in 2014 in the trend analysis of meropenem consumption, in the period 2012–2014, meropenem consumption decreased −0.08% per year (95%CI: −29.4 to 41.5) and in 2014–2019 decreased −15.5% per year (95%CI: −26.2 to −3.3). There was a 42% decrease in the consumption of meropenem during the intervention period with respect to the years 2012–2014 (Rate Ratio 0.58; 95%CI: 0.5–0.6); and increased consumption of ertapenem (RR 1.2; 95%CI 1.1–1.3) (Figure 1).

### 2.3. Impact on Resistance

The evolution of total consumption and by class of antibiotics is shown in Figure 2. Total consumption did not decrease and the consumption of antibiotics used as an alternative to carbapenem treatment increased: cefepime (RR 2; 95%CI: 1.8–2.3) (11.7 defined daily doses (DDD)/1000 occupied bed days (OBDs) in 2012–2014 vs. 18.5 in 2015–2019), piperacillin-tazobactam (RR 1.17; 95%CI: 1.1–1.24) (54.4 DDD/1000 OBDs in 2012–2014 vs. 63.8 in 2015–2019), aminoglycosides (RR 1.2; 95%CI: 1.1–1.3 (30.2 DDD/1000 OBDs in 2012–2014 vs. 36.2 in 2015–2019), and third-generation cephalosporins (59.4 DDD/1000 OBDs in 2012–2014 vs. 66.6 in 2015–2019), *p* < 0.001. Ciprofloxacin and metronidazole consumption remained similar, ciprofloxacin (88.5 in 2015–2019 vs. 89.8 DDD/1000 OBDs in 2012–2014), metronidazole (30.5 in 2015–2019 vs. 32.5 DDD/1000 OBDs in 2012–2014).

Global incidence of bacteremia adjusted by 1000 OBDs increased by 5.6% during the period 2015–2019 vs. 2012–2014. The incidence density of candidemia and hospital-acquired multidrug-resistant (MDR) bloodstream infections (BSIs) decreased after ASP start-up in a matching with decrease in carbapenem consumption (Figure 1 and Figure 3). In 2015–2019 candidemia and hospital-acquired MDR BSIs rate was 0.08/1000 OBDs vs. 0.13 in 2012–2014 (RR 0.6; 95%CI: 0.4–0.9, *p* = 0.02) (candidemia RR 0.56, 95%CI: 0.3–1.04; MDR BSIs RR 0.7; 95%CI: 0.4–1.1). Conversely, the incidence density of hospital-acquired BSIs caused by non-MDR strains of the same microorganisms under study increased 25% during the intervention period (RR 1.2; 95%CI: 1.08–1.4, *p* = 0.03).

Patients with hospital-acquired candidemia and MDR BSIs had higher 30-day all-cause mortality than patients with non-MDR BSIs: 28.3% vs. 18.7%, *p* = 0.04. The incidence density of 30-day all-cause mortality for hospital-acquired candidemia and MDR BSIs decreased during the intervention period by 49%, without reaching statistical significance (0.02/1000 OBDs in 2015–2019 vs. 0.04 in 2012–2014; RR 0.61; 95%CI: 0.3–1.3). 30-day all-cause mortality rate for hospital-acquired bacteremia caused by non-MDR microorganisms increased over time, without statistical significance: 0.12/1000 patients-days in 2012–2014 to 0.14 in 2015–2019 (RR 1.15; 95%CI: 0.8–1.7).

Throughout the study period, neither carbapenemase-producing microorganisms nor Vancomycin-resistant *Enterococcus* spp. were observed in hospital-acquired bacteremia, and the incidence of *Clostridioides difficile*-associated diarrhea remained stable (0.20/1000 OBDs in 2015 to 0.15 in 2019; RR 0.71, 95%CI: 0.4–1.1), *p* = 0.19. The resistance to piperacillin-tazobactam, cefepime, and aminoglycosides in *Klebsiella pneumoniae* and *Pseudomonas aeruginosa* isolated in blood cultures did not change significantly during the intervention period. Resistance to piperacillin–tazobactam in *Klebsiella pneumoniae* 8.4 ± 9.2 in 2012–2104 vs. 11 ± 6.8 in 2015–2019 and in *Pseudomonas aeruginosa* 16.6 ± 9.3 in 2012–2104 vs. 12.7 ± 12.1 in 2015–2019. Resistance to cefepime in *Klebsiella pneumoniae* 11.5 ± 6.8 in 2012–2104 vs. 6.7 ± 4.7 in 2015–2019 and in *Pseudomonas aeruginosa* 16.6 ± 9.3 in 2012–2104 vs. 12.7 ± 12.1 in 2015–2019. Resistance to gentamicin in *Klebsiella pneumoniae* 4.4 ± 6 in 2012–2104 vs. 6.9 ± 4.9 in 2015–2019 and resistance in *Pseudomonas aeruginosa* to tobramycin 17.9 ± 14.2 in 2012–2104 vs. 3.6 ± 4.9 in 2015–2019 and to amikacin 0% throughout the 2012–2019 period.

We monitored some indicators to assess if there were changes in hospital activity that could have contributed to the decrease in the incidence density of hospital-acquired candidemia or MDR BSIs. In the period 2015–2019 compared to 2012–2014 there were increases in: the number of blood cultures performed per 1000 OBDs (RR 2.4; 95%CI: 2.39–2.5), catheter-associated BSIs rate (RR 1.2; 95%CI: 1.01–1.5) and the number of surgical procedures (RR 2.7; 95%CI: 2.69–2.8). The consumption of parenteral nutrition was similar (0 35.8 units/1000 OBDs in 2012–2014 vs. 35.4 in 2015–2019) and antifungal consumption decreased by 18% (0 27.6 in 2015–2019 vs. 32.8 DDD/1000 OBDs in 2012–2014; RR 0.82; 95%CI: 0.76–0.88, *p* < 0.001), Table 3.

Alcohol-based hand-rub consumption increased progressively from 2015 to 2019 about 3.6% per year (average consumption 15.1 L/1000 OBDs). Overall antibiotic use in the hospital during the study years increased from 95 DDD/1000 OBDs in 2012–2014 to 106 in 2015–2019.

## 3. Discussion

The implementation of our ASP improved the prescription of carbapenems and decreased their consumption, without negative impact on patient safety. The acceptance of the intervention made by infectious diseases physician decreased days of treatment, inpatients days, and the incidence of candidemia and hospital-acquired BSIs caused by MDR bacteria, despite having increased somewhat the consumption of other antibiotics that have a lower ecological impact.

The care of patients with suspected infections is complex and metrics to assess ASP impact are poorly defined [20,21]. Though shorter in their extension, other studies reported briefer hospital stays with no difference in mortality [8,22], or reported decrease in antimicrobial resistance patterns [9,23]. There are no prospective studies outside the ICUs that analyze the development of infection caused by yeast during antibiotic treatment, and only two prospective studies refer to readmissions after discharging, without differences between patients with or without acceptance of ASP recommendation [24,25]. A study carried out in a carbapenem-resistant *Klebsiella pneumoniae* endemic hospital showed a decrease in antibiotics consumption without changes in candidemia or consumption of antifungal [26].

The incidence of *Clostridioides difficile*-associated diarrhea in our hospital is low and remained stable, and our results show a decrease in the incidence of hospital-acquired candidemia and MDR BSIs; this decrease was parallel to the decrease in meropenem consumption and to the decrease in the days of all antibiotic treatment. Candidemia is the fourth cause (4.9%) of hospital-acquired BSIs in our hospital. The decrease in the duration of all antibiotic treatments, the increase in de-escalation from carbapenems to narrower-spectrum antibiotics and the lesser development of colonization caused by yeast in patients in whom the intervention was accepted, have all undoubtedly contributed to the decrease in candidemia and the consumption of antifungals, without having carried out a direct intervention on the adequate use of antifungals as in other hospitals [27]. The decrease in the incidence of yeast infection has occurred despite the increase in the number of surgical interventions and the increase of intravascular catheter-associated BSIs, with similar use of parenteral nutrition throughout the study period [28]. Our results are reinforced by a recent study that shows, on a multivariate analysis, that the treatment of patients with carbapenems was associated with candidemia, adding weight to antimicrobial stewardship efforts and restriction of the use of these broad-spectrum antibiotics [15].

The antimicrobial stewardship programs are underfunded and it is necessary to prioritize those interventions that may have a greater impact [29,30]. We decided to follow the use of carbapenems because they are the antibiotics with the broadest antibacterial spectrum and with a rapid induction of beta-lactamases. We have decreased the incidence of candidemia and hospital-acquired MDR BSIs, and associated mortality, despite the increases in the total incidence of bacteremia and global consumption of antibiotics.

Understanding the relationship between antibiotic use and antibiotic resistance is therefore critical for the design of a rational antibiotic stewardship strategy. Because antibiotic use is uneven, total use does not distinguish between broad use-many people receiving a few prescriptions- and intensive use, a few people receiving many prescriptions [31]. It is necessary to focus on those antibiotics with the most ecological impact to try to reach an appropriate level of antibiotic use [32,33]. The decrease in the consumption of meropenem was associated with a slight increase in the consumption of piperacillin-tazobactam and cefepime, which may have less ecological impact as they are less AmpC inducers [34,35], and a slight increase in third-generation cephalosporins, aminoglycosides and ertapenem. The increase in ertapenem consumption may be due to its use to treat ESBL-producing *Enterobacteriaceae* infections, at the patient’s hospital discharge.

The results obtained in our study are undoubtedly due to the good acceptance of the interventions by the prescribers, higher than 42.3% for persuasive interventions described in the literature [36]. The intervention rejection level it seemed to be more in relation with clinicians’ attitudes in different hospitalization units and is not associated with the severity or comorbidity of the patient, according to our previously published results [18]. The decrease in meropenem consumption does not appear to be related to demographic changes and it is not justified by a decrease in the indications for use, because the incidence of sepsis and ESBL infections has not decreased, nor does it appear to be related to changes in the information of pharmaceutical companies, since the use of new antibiotics did not increase.

The strength of our study is the large number of variables analyzed and prospective data collection over 5 years to evaluate the impact of ASP, in a community hospital (medium size) without endemicity of carbapenemase-producing *Enterobacteriaceae*. We assessed compliance with local guidelines as the standard for appropriate therapy to reduce the more subjective method of expert opinion-based definitions. We implement prospective audit with intervention and feedback, which is the mainstay of antimicrobial administration in the patient setting and is currently recommended based on evidence largely generated in studies conducted in large tertiary care hospitals. Community hospitals provide most of the medical care in some countries, but due to their characteristics and limited resources, there are few long-term studies in these centers on auditing and feedback of carbapenem administration [37,38]. Our study can be replicated in the hospitals where targeting a specific antibiotic is needed and an infectious diseases physician is available for intervention.

Our study has several limitations. The study was extended to the entire hospital except ICU, but the candidemia and MDR BSIs acquired in ICU between 2012and 2019 accounted for 6.6% of hospital-acquired bacteremia, without significant differences between pre and post intervention period, and we believe that the activity of this service has not influenced our results.

The sample size does not allow use an interrupted time series regression to provide good stability to the results obtained for all the variables analyzed, but they reflect the changes in consumption of meropenem and in the incidence of hospital-acquired candidemia and MDR BSIs, after starting the ASP. Meropenem consumption level drop from 50.5 DDD/1000 OBDs in 2014 to 35.2 in 2015 coinciding with the start-up of ASP, and consumption has remained lower throughout the intervention period. This change in trend seem to be due to our intervention and not to changes in healthcare during the study period. As an ecologic study, it depicts association and not causal relations, because antibiotic resistance is temporally dynamic. Further work is needed to distinguish between different causal pathways [39].

The single-center design limits the possibility of generalizing our results to other hospitals, and including preferred methods, such as control groups or randomization was impractical. Although the complexity of hospitals and infection prevention and control policies may be different, it should be noted that in last five years carbapenem-resistant *Klebsiella pneumoniae* and *Pseudomonas aeruginosa*, and MRSA is lower in our hospital than in the rest of the hospitals in our geographical area [40].

## 4. Materials and Methods

This study is part of a larger assessment and the detailed description of the procedure has already been previously reported elsewhere and is briefly described below.

At the end of 2014, a team of professionals was constituted for ASP implementation, local guidelines for empiric antibiotic treatment were developed, and between January 2015 and December 2019, a prospective follow-up of carbapenems use was performed. We analyzed the evolution of adequacy of carbapenems prescription to local guidelines and clinical impact, antibiotic consumption and the incidence of bloodstream infections acquired in the hospital.

A descriptive study pre-post-intervention was conducted between January 2012 and December 2019.

The study was conducted in a 350-bed teaching hospital without endemicity of carbapenemase-producing Enterobacteriaceae, located in Galicia (autonomous community of northwest of Spain). The hospital has one ICU with 10 beds and does not have transplant programs.

The infection prevention and control program was the same throughout the study. From 2012, an infectious diseases physician performed prospective active surveillance of all episodes of bloodstream infections (BSIs) [41].

Interventions: 

Patients who started treatment with carbapenems available in our hospital (meropenem and ertapenem) were identified every day throughout the 5 years using a drug dispensation program (all hospital units, except ICU). Prescriber counselling measures were performed the first day of prescription, and annual training on optimization of antibiotic use was carried out for the first 3 years, targeting trainee pharmacists and physicians. An annual antimicrobial stewardship program update was presented at a hospital general clinical session.

An infectious diseases physician was released 6 h a week to perform active surveillance. For each case, the electronic medical record was reviewed by infectious diseases physician and antibiotic treatment recommendations to prescribers were given, on a face-to-face or telephone conversation basis, or through an electronic medical record. Additional differential diagnoses, investigations, and adjunctive therapy (for example, removal of urinary or central venous catheters, and drainage of infected collections) were also recommended. Adherence to, or rejection of the recommendations were reviewed by an infectious diseases physician 24 and 48 h post-recommendation, as part of the ASP workflow. Data were obtained by monitoring the information recorded in the electronic medical record. Prospective and protocolized information was collected for each case [18]. This study was approved by the Institutional Review Boards and Ethics Committee.

### 4.1. Adequacy of Treatment

Appropriate treatment with carbapenems was considered when it was prescribed in patients with: (1) severe sepsis [42]; (2) history of ESBLs colonization; or (3) hospital-acquired infection in which a broad-spectrum antibiotic treatment was considered necessary.

### 4.2. Clinical Impact

In cases in which carbapenem treatment was not justified during 2015–2019, a comparison was made between the cases with accepted intervention and cases with rejected intervention in their clinical outcome, days of antibiotic treatment, collateral damage (development of phlebitis, resistance to treatment, diarrhea caused by *C. difficile*, or for colonization-infection with *Candida* spp.), inpatient days, and hospital readmission. All-cause mortality was defined over one month of follow-up; 30-day infection-related and all-cause readmission were defined as readmission occurring within 30 days after discharge from current admission. During the study period, antibiotic consumption was assessed as defined daily doses (DDD) per 1000 occupied bed days (OBDs) [43]. The impact on the use of antibiotics was made comparing the DDD/1000 OBDs between the years 2012 and 2019.

### 4.3. Impact on Resistance

We analyzed, between January 2012 and December 2019, the evolution of incidence density per 1000 OBDs of hospital-acquired BSIs produced by the most frequently isolated bacteria and by *Candida* spp.

Hospital-acquired BSIs were defined as those diagnosed from blood cultures obtained ≥48 h after hospital admission or in those cases when, even occurring in the first 48 h, the patient had been hospitalized during the previous two weeks.

The identification of blood isolates and the determination of resistance to antibiotics were performed according to Clinical Laboratory Standard International (CLSI). The MDR categorization was applied for ESBLs or carbapenemase-producing Enterobacteriaceae, all isolates of methicillin-resistant *Staphylococcus aureus* (MRSA), and all *Pseudomonas aeruginosa* and *Acinetobacter baumannii* strains fulfilling the German Society for Hygiene and Microbiology criteria for MDR organisms [44]. Colonization was defined as the isolation of the organism from a non-sterile site in the absence of symptoms, and infection when the patient’s doctor prescribed treatment.

### 4.4. Statistical Analysis

A descriptive and comparative study of the variables was performed. Quantitative variables are reported as means ± standard deviations, and categorical as frequencies (%). Variables were compared between groups using Chi-square test or Fisher exact test for categorical variables, Student t-test, or Mann–Whitney U for continuous variables, as appropriate. Associations between the variables were expressed as odds ratios (ORs) and 95% confidence intervals (CIs). Antibiotic consumption (DDD per 1000 occupied bed days), resistance rates per 1000 OBDs of hospital acquired BSIs with a 95%CI and rates of mortality were calculated as Poisson event rates, and compared by testing for homogeneity of rates. Statistical analysis was performed using SPSS software version 19. All tests were 2-tailed; a *p* value < 0.05 was regarded as statistically significant.

An interrupted time-series analysis was performed to verify any significant change in antibiotic consumption line; we used the Joinpoint Regression Program 4.5.0.1 to run the calculations.

## 5. Conclusions

The results of this study show that the decrease and better use of carbapenems achieved by our stewardship program could have a sustained clinical and ecological impact over five years, reducing inpatient days and incidence of hospital-acquired MDR BSIs and candidemia, despite the increase in consumption of other antibiotics with less impact on the microbiome.

## Figures and Tables

**Figure 1 antibiotics-10-00015-f001:**
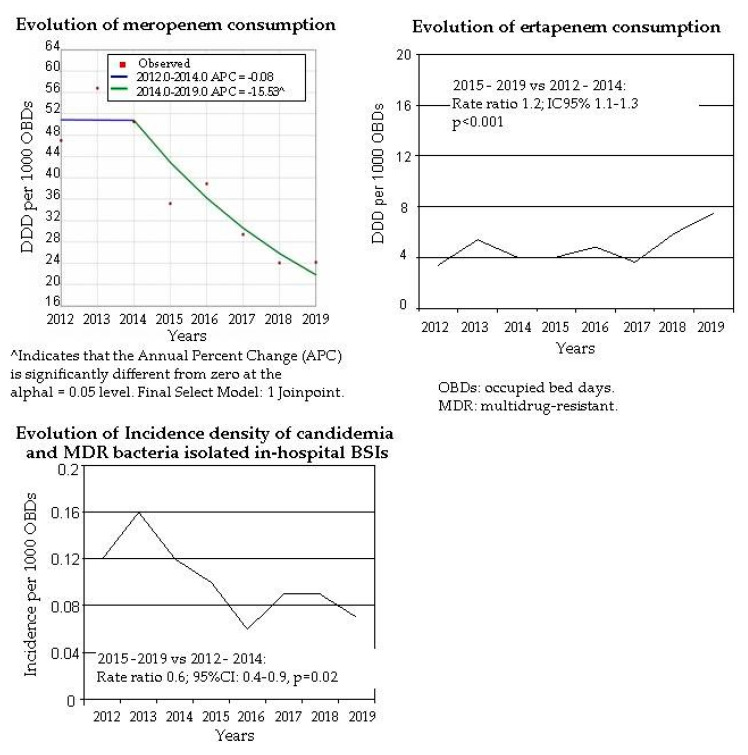
Evolution of carbapenems consumption and incidence of candidemia and MDR bacteria isolated in-hospital bloodstream infections (BSIs).

**Figure 2 antibiotics-10-00015-f002:**
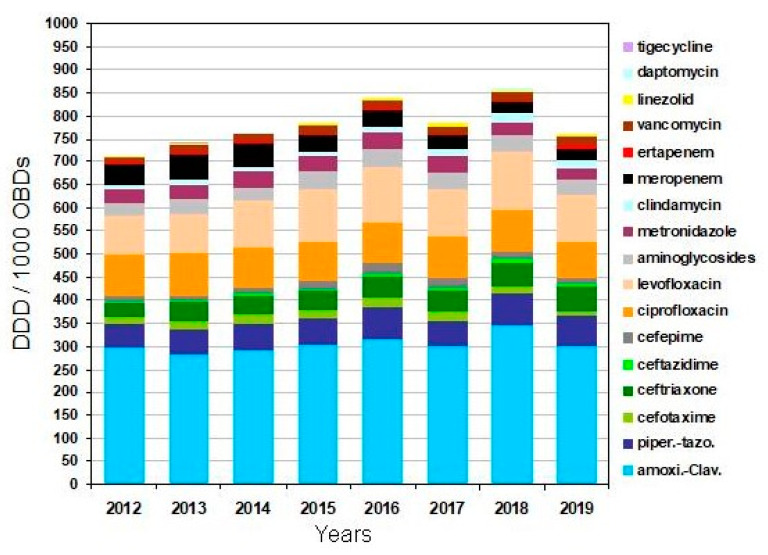
Evolution of antibiotic consumption between the years 2012 to 2019.

**Figure 3 antibiotics-10-00015-f003:**
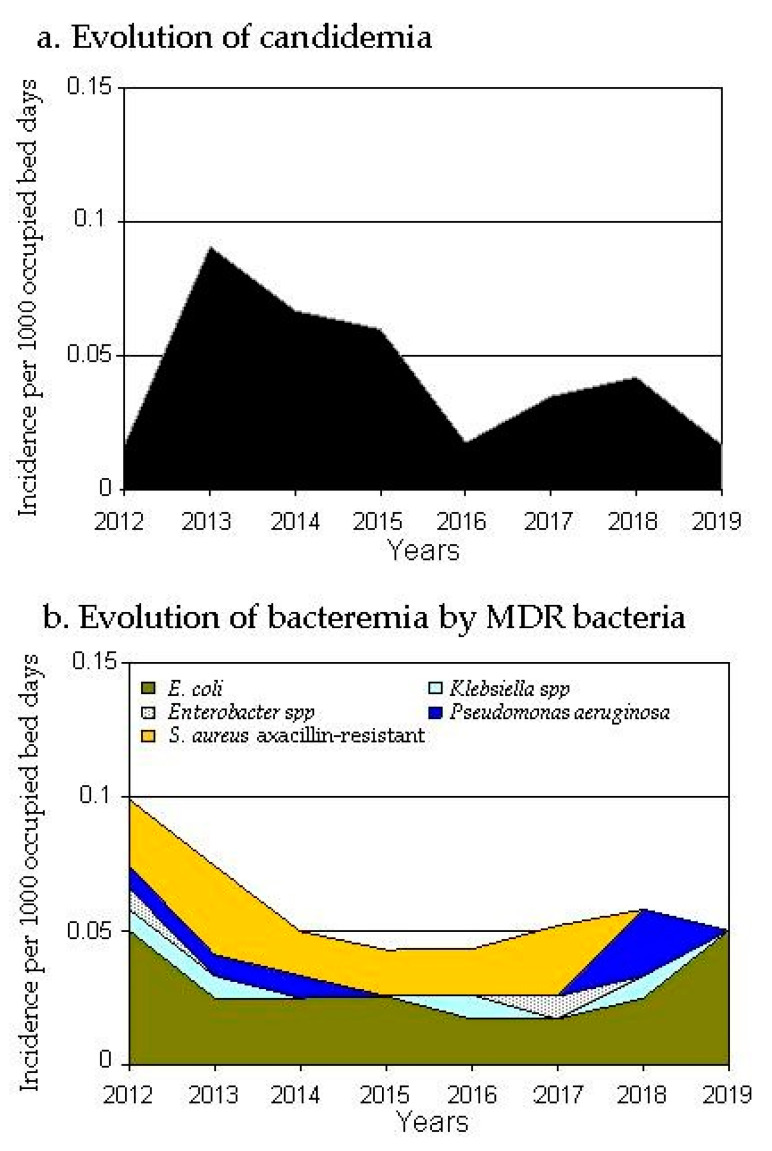
Evolution of the incidence density of candidemia and MDR bacteria isolated most frequently in-hospital bloodstream infections.

**Table 1 antibiotics-10-00015-t001:** Baseline demographic and clinical characteristics of patients with and without acceptance of ASP Recommendations. Years 2015–2019.

Variables	Intervention Accepted n = 371	Intervention Rejected n = 77	*p*
Male gender, n (%)	243 (65.5)	48 (62.3)	0.6
Median age ± SD, years (range)	67.2 ± 15.5 (10–96)	66 ± 18.8 (6–95)	0.6
Charlson comorbidity score, Median ± SD, (range)	5.2 ± 2.9 (0–13.6)	4.8 ± 2.9 (0–12)	0.4
Neutropenia, <500/mL	7 (1.9)	3 (3.9)	0.4
Sepsis	15 (4)	5 (6.5)	0.4
Site of infections, n (%)			
Pulmonary	64 (17.3)	24 (31.2)	0.007
Abdominal	99 (26.7)	36 (46.8)	0.001
Skin/soft tissue	25 (6.7)	1 (1.3)	0.06
Urinary	131 (35.3)	12 (15.6)	0.001
Other	52 (14)	4 (5.2)	0.04
Acquisition place of infection			
Hospital onset	127 (34.2)	33 (42.8)	0.15
Healthcare-associated	134 (36.1)	28 (36.4)	1
Community-associated	110 (29.6)	16 (20.8)	0.13

**Table 2 antibiotics-10-00015-t002:** Clinical results of patients with and without Acceptance of ASP recommendations. Years 2015–2019.

Variables	Intervention Accepted n = 371	Intervention Rejected n = 77	*p*
Evolution to healing	329 (88.7%)	64 (83.1%)	0.4
Death caused by infection	15 (4%)	7 (9.1%)	0.08
All-cause crude death	42 (11.3%)	12 (15.6%)	0.3
Readmission in a month	15 ^#^ (4%)	4 * (5.2%)	0.6
Adverse effects	42 (11.3%)	7 (9.1%)	0.8
Phlebitis	64 (17.3%)	11 (14.3%)	0.6
Development of resistance to treatment	8 (2.2%)	0 (0%)	0.4
Diarrhea caused by *C. difficile*	6 (1.6%)	2 (2.6%)	0.6
Colonization-Infection with *Candida* spp.	40 (10.8%)	11 (14.3%)	0.4
Days of antibiotic treatment (intervention series)	11 ± 10.2	12.7 ± 8.5	0.2
Total inpatient days, X ± SD	17.7 ± 16.7	25.3 ± 22.3	0.006
Inpatient days post-intervention, X ± SD	12.5 ± 14.2	16.7 ± 18.9	0.03

^#^ 9 relapses of the infection, 6 due to other causes. * No relapse of infection due to other cause.

**Table 3 antibiotics-10-00015-t003:** Potential Changes in Healthcare during the Study Period by Year.

Healthcare Variable.	2012	2013	2014	2015	2016	2017	2018	2019
Nº of patients admitted	14,721	14,615	14,979	14,867	14,852	15,248	15,561	15,641
Nº of inpatient days	119,885	121,181	119,615	116,588	114,072	114,864	120,133	119,350
Blood cultures performed, No.	3242	3340	2985	3003	3419	3074	3554	3331
Nº blood cultures/1000 OBDs *	27.0	27.6	24.9	25.8	29.9	26.8	29.6	27.9
Hospital-acquired BSIs per 1000 OBDs *	0.8	0.9	0.8	0.8	0.8	1.0	1.4	1.0
Hospital-acquired no-MDR BSIs/1000 OBDs *	0.7	0.8	0.7	0.7	0.8	0.9	1.3	0.9
Intravascular catheter-associated BSIs/1000 OBDs *	0.3	0.4	0.5	0.4	0.4	0.5	0.7	0.4
Surgical procedures, Nº/1000 OBDs *.	74	77	74	78	82	80	76	79
Case mix index ^+^	1.5	1.5	1.6	1.6	0.8	0.9	0.9	0.9
Parenteral nutrition units, No./1000 OBDs	29.1	38.7	39.5	46.1	41.3	34.0	28.9	26.5
Consumption of antifungals, DDD/1000 OBDs *	29.1	38.4	30.8	34.9	33.1	25.5	23.6	21.1

MDR: multidrug-resistant. BSIs: bloodstream infections. OBDs: occupied bed days. * In the period 2015–2019 compared to 2012–2014 increased the number of blood cultures performed, hospital-acquired BSIs and hospital-acquired no-MDR BSIs, catheter-associated BSIs and surgical procedures; antifungal consumption decreased (*p* < 0.001). ^+^ The calculation system was modified after 2015 (Ref. [19]), case mix index increased progressively between 2012 and 2014 and between 2015 and 2019.

## Data Availability

Raw data is available to the public, upon request to the authors.

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
