# Peer review of "Long-Term Carbapenems Antimicrobial Stewardship Program"

_antibiotics, 2020, doi:10.3390/antibiotics10010015_

Round 1

Reviewer 1 Report

In general a very well written study that shows that a implementation of a action plan and a better control of usage of antibiotics can lead to a decrease of certain drugs.

General comments: 1. The usage of capital letters for words such as “Stewardship Programs” should be limited to when it is a name of a intervention or similar.
2.In the PDF there are a lot of unnecessary space between words and numbers. 3. The study have been very focussed on ESBL and drugs such as Menopenem and erthapenem. To really make a point in the scientific world more drugs should have been taken into account especially when the decrease seems to be long term and relatively rappid as presented. ( are there other drugs use instead and how is the resistant levels for these, my guess increasing…this is more clear in the abstract then in the real manuscript so why not shown in results in a clear way? 4. If you have a intervention like this, You need to show it in the resultsection as well so the results can be discussed fro the intervention not only the fussy part in methods.

When I do statistics from the data presented I get slightly other numbers please check again. Also since there are more men in the statistics this needs an explination in discussion.I also suggest to not have both +- and percentage with more than one decimal

I think the authors are bold in these 3 paragraphs and suggest a rewriting to better agree with the rest of the article” The results obtained in our study are undoubtedly due to the good acceptance of the interventions by the prescribers, higher than the median change in antibiotic prescribing (42.3%) for the persuasive interventions described in the literature [38]. The intervention rejection level was not associated with the severity or comorbidity of the patient and it seemed to be more related to clinicians’ attitudes in different hospitalization units [17].

The strength of our study is the large number of variables analyzed and prospective data collection over 5 years to evaluate the impact of ASP, in a hospital without endemicity of carbapenemase-producing Enterobacteriaceae. We assessed compliance with local guidelines as the standard for appropriate therapy to reduce the more subjective method of expert opinion-based definitions. This study can be replicated in settings where targeting a specific antibiotic is needed and a ID physician is available for intervention.

Our study has several limitations. The study was restricted to those wards with electronic medication dispensing system (all the hospital except ICU), but the MDR BSIs acquired in ICU between 2012-2019 accounted for 6.6% of hospital-acquired bacteraemias, without significant differences between pre and post intervention period, and we believe that the activity of this service has not influenced our results.”

This since who is the prescriber and the pharmacist needs to work as a team and the results only show in a bias way that the prescriber have changed, this could have been done due to new info on market, info from pharmaceutical companies etc, influencing the result. Strength is the numbers in the study and that you have collected data. The replication is doubtful.

Author Response

We welcome your valuable comments on our manuscript

1. We have tried to improve the use of capital letters for words.

We have improved the introduction. We have included in the lines 56-61 “In addition to the impact on bacterial resistance, different antimicrobial agents have a different impact on the human gastrointestinal colonization by Candida species, depending on the pharmacological and antibacterial properties of each drug. It has been demonstrated in murine model that carbapenems cause high increase of yeast counts in the gastrointestinal tract [14], and the use of carbapenems in the treatment of hospitalized patients is associated with the development of candidemia [15].”

We have tried to improve the description of methods, line 265, lines 272-275 and lines 279-281.

We have improved the conclusions trying that they are supported by the results.

2. We have tried to correct unnecessary space between words and numbers. It may be due to a problem generating the PDF at the publisher.

3. We have added in figure 2 the evolution of the total consumption and by class of antibiotics between the years 2012-2019.

In the lines 113-121 we include information on the use of other drugs instead of carbapenems and in the lines 144-150 on the evolution of resistance for these.

4. You can see our answer in point 3.

    We have checked the data again.

    We have tried not to use more than one decimal.

    We have rewritten these paragraphs.

    In the lines 215-223 we include  “The intervention rejection level it seemed to be more in relation with clinicians’ attitudes in different hospitalization units and is not associated with the severity or comorbidity of the patient, according to our previously published results [18]. The decrease in the use of Meropenem does not appear to be related to demographic changes and it's not justified by a decrease in the indications for use, because the incidence of sepsis and ESBL infections has not decreased, nor does it appear to be related to changes in the information of pharmaceutical companies, since the use of new antibiotics did not increase.”

    In lines 224-234  we include “The strength of our study is the large number of variables analyzed and prospective data collection over 5 years to evaluate the impact of ASP, in a community hospital (medium size) without endemicity of carbapenemase-producing Enterobacteriaceae. We assessed compliance with local guidelines as the standard for appropriate therapy to reduce the more subjective method of expert opinion-based definitions. We implement prospective audit with intervention and feedback, which is the mainstay of antimicrobial administration in the patient setting and is currently recommended based on evidence largely generated in studies conducted in large tertiary care hospitals. Community hospitals provide most of the medical care in some countries, but due to their characteristics and limited resources, there are few long-term studies in these centers on auditing and feedback of carbapenem administration [41,42]. Our study can be replicated in these hospitals where targeting a specific antibiotic is needed and a ID physician is available for intervention.

Reviewer 2 Report

García-Rodríguez et. al. review the carbapenem usage in a single hospital setting from 2012 to 2019 and report that adequacy of carbapenem usage increased by about 60% in 2019 compared to that of 2015. This decrease in carbapenem usage has resulted in lower in-patient days and hospital acquired bloodstream infections, despite increased antibiotic consumption.

Major concern:

The overall article is well written, and as the author states, it has several limitations, including the study involved a single hospital. The article is limited in conclusion and readership. If the authors can review and discuss at least a couple of such single hospital studies that helped in antimicrobial stewardship, can help in broader applicability of this work.

Minor:

Some short-form acronym that appeared that were not described before in full needs to abbreviate in full.

Author Response

We welcome your valuable comments on our manuscript.

     We have improved the introduction. We have included in the lines 56-61 “In addition to the impact on bacterial resistance, different antimicrobial agents have a different impact on the human gastrointestinal colonization by Candida species, depending on the pharmacological and antibacterial properties of each drug. It has been demonstrated in murine model that carbapenems cause high increase of yeast counts in the gastrointestinal tract [14], and the use of carbapenems in the treatment of hospitalized patients is associated with the development of candidemia [15].”

    We have tried to improve the description of methods, line 265, lines 272-275 and lines 279-281.

    We have tried to improve the presentation of the results. We have added in figure 2 the evolution of the total consumption and by class of antibiotics between the years 2012-2019.

    In the lines 113-21 we include information on the use of other drugs instead of carbapenems and in the lines 144-150  on the evolution of resistance for these.

    We have improved the conclusions trying that they are supported by the results.

Major concern:

    In the lines 228-234 we include “We implement prospective audit with intervention and feedback, which is the mainstay of antimicrobial administration in the patient setting and is currently recommended based on evidence largely generated in studies conducted in large tertiary care hospitals. Community hospitals provide most of the medical care in some countries, but due to their characteristics and limited resources, there are few long-term studies in these centers on auditing and feedback of carbapenem administration [41,42]. Our study can be replicated in these hospitals where targeting a specific antibiotic is needed and a ID physician is available for intervention.”

Minor:

    We have reviewed and corrected some short-form cronym that appeared without having been previously described in the text.

Reviewer 3 Report

García-Rodríguez et al. present a follow-on analysis of a previously published study wherein carbapenems were the focus of antimicrobial stewardship and the review period was extended. The authors analyzed antibiotic consumption patterns, clinical characteristics, and infection characteristics (e.g., BSI and fungal infection rates), and clinical outcomes according to whether or not the antimicrobial stewardship intervention was accepted or rejected. They found that duration of carbapenem use decreased and hospitalization duration decreased post-ASP intervention implementation. The authors noted reductions in antifungal consumption and a reduction in Candida BSI after ASP was introduced. The study is interesting because carbapenem use (by DDD/1000 OBD) appears to have been egregiously high for a rather small 350-bed hospital (~50 DDD/1000 OBD reduced to ~ 25 DDD/1000 OBD by the end of review).

The analysis is challenging because many ecologic factors were at play (e.g., per Figure 2, MDR BSI appeared to have been declining before the intervention while case-mix was rising). The analysis would be greatly improved by more granular data and a robust interrupted time series analysis. I have the following major comments and critiques.

1. The introduction focuses heavily on MDR bacterial pathogens and Cdiff but the findings of the paper are mostly related to fungal infections and antifungal use. Cdiff was flat and per Fig 2 it looks like MRSA and E coli were also flat. Thus, the intro can discuss the relationship between carbapenem use and fungal infections as well.

2. The authors have performed a segmented regression of the meropenem consumption data over years, but this should be performed on monthly data. Why do the author's not have monthly consumption data for their own hospital? The same can be said for BSI and other data expressed per annum rather than monthly. This would allow assessment of changes in both slope and level (intercept). Both are of interest as step changes show the immediate effect (if the intervention worked quickly) whereas slope changes show the change over time. It would appear that the joinpoint analysis fixes the intercept. The lack of data granularity reduces the ability to discern the step change. With more granular data, a more robust analysis (including time varying covariates) can be completed.

3. All antibiotic consumption should be expressed in 1000 OBD. Once consumption is expressed on the same scale as the other process measures like BSI, MDR BSI, surgery, and catheter BSI, etc. These measures can then be correlated. These correlations would support the author's presumption that ASP carbapenem interventions were related to these outcomes. Candida should be considered separately from other BSI because reductions in carbapenem consumption could be directly caused by independent reductions in MDR bacteria (correlation/causation problem). Candida infections deserve separate mention, see below.

4. As mentioned above, the correlations for carbapenem consumption and candida BSI and antifungal consumption should be considered separately. It is well known that carbapenem exposure increases colonization with yeast. PMID: 2146241 and 22680984. Also, carbapenem reciept has been classified as a risk factor for candida overgrowth clinically: PMID: 33185290 and 11561137.

5. Within the authors' center, case mix appeared to progress over over years (structural break in 2015 due to re-calculation) suggesting more complex patients were seen over time, although their rates of carbapenem resistance and MRSA were lower than comparator hospitals. If this is true, it would be important to adjust their analysis for the changes in demographics and case-mix, see comment 1 related to more robust analysis.

6. Details of the intervention are still somewhat unclear. I understand that prospective audit and feedback was a rolling intervention during the intervention period (6 hr/wk). However, training and education tend to have stochastic effects, so a timeline or graphical representation of when education was given relative to carbapenem consumption would be informative. Such a graphic would require monthly consumption data over the study period.

Minor comments:

Figure 1, germs is not an appropriate term. Prefer pathogens but see comment about separating candida above.

Author Response

We welcome your valuable comments on our manuscript.

1. We have improved the introduction and discussed the relationship between carbapenems and fungal infections. We have included some of the references that you have suggested.

    We have included in the lines 56-61 “In addition to the impact on bacterial resistance, different antimicrobial agents have a different impact on the human gastrointestinal colonization by Candida species, depending on the pharmacological and antibacterial properties of each drug. It has been demonstrated in murine model that carbapenems cause high increase of yeast counts in the gastrointestinal tract [14], and the use of carbapenems in the treatment of hospitalized patients is associated with the development of candidemia [15].”

2.  At this time we do not have data on monthly antibiotics consumption, due to the laboriousness of obtaining them and the small size of our hospital. We could give BSI data by months, but the small number, less than 10 in some months, does would not allow use an interrupted time series regression.

     In figure 1, the graph displays the pre-intervention trend of the annual consumption of Meropenem. The consumption level of Meropenem drop from 0.5 DDD / 1000 OBD in 2014 to 0.3 in 2015, and between the years 2015-2019 all the points are well below the hypothetical scenario in which the intervention has not been carried out, the counterfactual scenario.

3.  We have expressed all antibiotic consumption in 1000 OBD.

     Due to the small sample size, we have decided to consider Candida and other BSI together for the analysis, analyzing Candida and other BSI separately decreases the consistency of our results. In the lines 129-130 we include “[candidemia RR 0.56, 95%CI: 0.3-1.04; MDR BSIs RR 0.7; 95%CI: 0.4-1.1].

    In the lines 228-234 we include “The decrease in the use of Meropenem does not appear to be related to demographic changes and it's not justified by a decrease in the indications for use, because the incidence of sepsis and ESBL infections has not decreased, nor does it appear to be related to changes in the information of pharmaceutical companies, since the use of new antibiotics did not increase.”

    In the lines 195-198 we include “Our results are reinforced by a recent study that shows, on a multivariate analysis, that the treatment of patients with carbapenems was associated with candidemia, adding weight to antimicrobial stewardship efforts and restriction of the use of this broad-spectrum antibiotic [15].”

4. We have added comments on the association of carbapenems with candidemia in the introduction (lines 56-61) and discussion (lines 195-198) sections, and we have introduced some of the references that you suggest.

5. See our answer in point 1. There have been no changes in demography in our geographical environment, except that each year the number of inhabitants decreases somewhat and the mean age of the patients increases. The incidence of sepsis and ESBL infections has not decreased (two of the justified indications for treatment with carbapenems).

6.  In methods section we add more information about education training, lines 272-275 and 279-281.

Minor comments:

     We have changed germs to pathogens in figure 1.

Round 2

Reviewer 3 Report

I have read the revised version of the paper. Overall, the work is very sloppy both from a methodological and writing perspective. There are too many grammatical (e.g., lines 112 and 304: resistances should be resistance), usage (e.g., evolution should be outcome), and scientific errors (e.g., line 132 p value is missing the decimal) to fully enumerate completely. The manuscript needs extensive copy editing for clarity and readability.

A major scientific error is that the calculation of DDD/1000 days is wrong for every instance shown. If there were 5 DDD/100 OBD in the prior version of the paper, then it should be 50 DDD/1000 OBD now. The authors have reversed this calculation and shown us DDD/ 10 OBD now.

Regarding consumption, I find it unbelievable that they cannot obtain monthly consumption data since it is just purchase data...hospitals track this information. Ask your pharmacy department for purchase records aggregated monthly. Nevertheless, it is acknowledged that the trend is down. Crucially, however, the authors cannot estimate the difference between their program's immediate vs. long term effect with any precision (attempted in lines 243-244), which defeats the purpose of the paper. This is a major barrier to causal inference. To be clear, more granular data would strengthen the temporal relationship. 

Line 95, the reduction in yeast colonization was numerically lower but not statistically significant. Please be clear about this.

Line 142-151, percentages for resistance are missing for all use instances.

Line 313, The MDR BSI definition is unclear as all candida are lumped together. C. albicans is not MDR organism. Not all candida spp are MDR...for example I would be surprised if they had any C. auris in their center. It is academically questionable to combine these disparate organisms merely to increase statistical power. A simple solution is to analyze these separately throughout the paper.

Author Response

I have read the revised version of the paper. Overall, the work is very sloppy both from a methodological and writing perspective. There are too many grammatical (e.g., lines 112 and 304: resistances should be resistance), usage (e.g., evolution should be outcome), and scientific errors (e.g., line 132 p value is missing the decimal) to fully enumerate completely. The manuscript needs extensive copy editing for clarity and readability.

     We have proceeded to review the writing of the article.

     We have corrected grammatical (e.g., resistance, lines 112 and 310), usage (e.g., evolution should be outcome), and scientific errors (e.g., line 132 p value with the decimal).

A major scientific error is that the calculation of DDD/1000 days is wrong for every instance shown. If there were 5 DDD/100 OBD in the prior version of the paper, then it should be 50 DDD/1000 OBD now. The authors have reversed this calculation and shown us DDD/ 10 OBD now.

     We detect our error when writing the consumption data in DDD / 1000 OBD and when we communicate it to the editor, the revised version of the document had already been sent to you. We have now corrected our scientific error in the text and in the figures.

Regarding consumption, I find it unbelievable that they cannot obtain monthly consumption data since it is just purchase data...hospitals track this information. Ask your pharmacy department for purchase records aggregated monthly. Nevertheless, it is acknowledged that the trend is down. Crucially, however, the authors cannot estimate the difference between their program's immediate vs. long term effect with any precision (attempted in lines 243-244), which defeats the purpose of the paper. This is a major barrier to causal inference. To be clear, more granular data would strengthen the temporal relationship. 

     Regarding consumption, MEDICINES CONSUMPTION DATA is not purchase data. Antimicrobial consumption data was provided by Pharmacy department, and the consumption unit measure internationally recognized is DDD / 100 occupied bed days (OBDs) or DDD / 1000 inhabitants and per day.

     The European Surveillance of Antimicrobial Consumption Network (ESAC-Net) of the European Centre for Disease Prevention and Control (ECDC) collects reference data on the consumption of antimicrobials for systemic use in the community and in the hospital sector in EU expressed as a number of WHO Defined Daily Doses (DDD) per 1000 inhabitants and per day (see https://www.ecdc.europa.eu/en/about-us/partnerships-and-networks/disease-and-laboratory-networks/esac-net). WHO Defined Daily Doses (DDD) are defined and periodically reviewed by the WHO Collaborating Centre for Drug Statistics Methodology (se https://www.whocc.no/atc_ddd_index/).

     DDD / 1000 OBDs = consumption of any antimicrobial expressed in grams/DDD x 1000/OBDs.

    To calculate the DDD/1000 OBDs of any antimicrobial, first we must multiply the consumed units (vials, tablets,.. .) by the grams contained in each vial, tablet,… ; then divide by the WHO Defined Doses (DDD) and OBDs in a definite period of time and multiply by 1000.

     Thus we can make consumption comparations between different hospital sizes or countries, trends by country, geographical distribution….

     We have now expressed all antibiotic consumption in 1000 OBD

Line 95, the reduction in yeast colonization was numerically lower but not statistically significant. Please be clear about this.

     In line 95-96 we clarify that the reduction in yeast colonization is not statistically significant.

Line 142-151, percentages for resistance are missing for all use instances.

     In lines 145-153 we add data on the evolution of  resistance to antibiotics that increased their consumption, in hospital-acquired BSIs bacteria most frequently isolated.

Line 313, The MDR BSI definition is unclear as all candida are lumped together. C. albicans is not MDR organism. Not all candida spp are MDR...for example I would be surprised if they had any C. auris in their center. It is academically questionable to combine these disparate organisms merely to increase statistical power. A simple solution is to analyze these separately throughout the paper.

     We have tried to analyze candidemia and MDR BSI separately, in all the text and figures, lines 26, 129-130, 135-139, 155-157, 176-177, 189-190, 206-207, 240-242, 246-247, 311-313, 317-321, figure 1 and figure 3.